# Sustainable Irrigation System for Farming Supported by Machine Learning and Real-Time Sensor Data [note 1]

**DOI:** 10.3390/s21093079

**Published:** 2021-04-28

**Authors:** André Glória, João Cardoso, Pedro Sebastião

**Affiliations:** 1Department of Science, Information and Technology, Instituto Universitário de Lisboa (ISCTE-IUL), 1649-026 Lisbon, Portugal; jmbco1@iscte-iul.pt (J.C.); pedro.sebastiao@iscte-iul.pt (P.S.); 2Instituto de Telecomunições (IT), 1049-001 Lisbon, Portugal

**Keywords:** Internet of Things, machine learning, wireless sensor networks, sustainable farming, sustainability, water efficiency

## Abstract

Presently, saving natural resources is increasingly a concern, and water scarcity is a fact that has been occurring in more areas of the globe. One of the main strategies used to counter this trend is the use of new technologies. On this topic, the Internet of Things has been highlighted, with these solutions being characterized by offering robustness and simplicity, while being low cost. This paper presents the study and development of an automatic irrigation control system for agricultural fields. The developed solution had a wireless sensors and actuators network, a mobile application that offers the user the capability of consulting not only the data collected in real time but also their history and also act in accordance with the data it analyses. To adapt the water management, Machine Learning algorithms were studied to predict the best time of day for water administration. Of the studied algorithms (Decision Trees, Random Forest, Neural Networks, and Support Vectors Machines) the one that obtained the best results was Random Forest, presenting an accuracy of 84.6%. Besides the ML solution, a method was also developed to calculate the amount of water needed to manage the fields under analysis. Through the implementation of the system it was possible to realize that the developed solution is effective and can achieve up to 60% of water savings.

## 1. Introduction

Presently, as has been true for the past few centuries, agriculture is the main supplier of food for society, with more than 74% of the daily consumption of populations coming from agricultural fields [1]. Not only is agriculture essential to the population, it is also one of the major activities in terms of water consumption, with more than 70% of the world fresh water being used for these activities. In addition, to keep production for the growing number of consumers and with water scarcity being a worldwide concern, it cannot be accepted that still 30% of that water is wasted or misused due to many situations such as lack of control, according to [2]. It is to mitigate these problems that sustainability and the use of technology in agriculture have become big trends.

With the evolution of technology and the way information is used in our daily lives and activities, it is now possible to know any kind of information at any time and in any place with a simple smartphone or computer; devices and technology are being implemented in activities that until now have never been used. This evolution is possible due to the great number of small and cheap devices that are connected to the Internet and are capable of collecting crucial information. These devices can also combine a wide range of technologies, leading to the creation of the Internet of Things (IoT) [3].

The proliferation and success of IoT solutions was due to the ability of creating networks of sensors and actuators, technology that had a big evolution in recent years, with the creation of Wireless Sensor Networks (WSN) that, when working in parallel with IoT, can provide the user a numerous amount of functionality and solutions, allowing connecting a large number of IoT devices, send data and control information through online platforms or mobile apps [4].

Agriculture depends on water to achieve a successful harvest, with a need to administrate the correct amount of water to guarantee the health and quality of the fields. With this water management still being done by humans, mostly in trial and error fashion, that leads to complex and inaccurate results. Using sensors and real time information allows for a correct use of water, not only saving this natural resource that is more and more scarce these days, but also creating a more sustainable way of farming, with better conditions and fewer costs to the farmer [5].

Thus, this paper presents a system capable of improving the sustainability and efficiency of irrigation in agricultural fields, that is autonomously adapted, using artificial intelligence and machine learning, in order to work according to the field needs. Through the collection of real time local data directly in the field, it is possible to improve the irrigation system, as well as give the owner the appropriate information, such as the best time of the day for irrigation. This system allows the farmer to have a better understanding of their fields and reduce the costs of water and maintenance.

The system uses a wide range of sensors that are strategically spread over the agricultural fields in order to collect the data needed for the correct monitoring, using a Wireless Sensor Network (WSN). Then this data is sent, using NB-IoT to a cloud server, where it is stored and analyzed, with machine learning techniques, to decide the best actions to perform in the field.

Besides the system case study, the goal for this paper is to understand which are the best practices for this type of systems, perform a machine learning study to understand the best algorithm to use for irrigation scheduling based on sensor data and understanding if a machine learning approach provides improvements facing typical or smart irrigation solutions. These objectives are outlined with the numerous case studies already found in the literature, each using its methodology or algorithm, but none providing a complete system that includes the best solutions.

After this introduction, the highlights of the developments already existent in the literature are presented, including the use of WSN and machine learning in agriculture, as well as our previous work. It follows with a detailed description of the system architecture, including all the hardware, software and learning systems. A study of Machine Learning techniques, that includes the training methodology and a comprehensive comparison of these techniques for irrigation purposes will be shown. A real case implementation scenario is described, followed by the results and discussion. Finally, the conclusions for the system architecture, Machine Learning study and implementation results are presented.

## 2. Related Work

With the evolution of technology and the constant development of solutions in the area of IoT, in parallel with intelligent solutions produced in the area of artificial intelligence and machine learning, multiple solutions have already been developed with the aim of combining the two in order to obtain cheaper solutions and with the purpose of saving natural resources.

There is an excellent literature review [6] on the application of machine learning in the whole spectrum of agriculture, from seeding, harvest, pest and diseases control, quality checks and all the other steps involved in the supply chain of agriculture. Regarding irrigation, the authors pointed that irrigation scheduling is among the most important roles in agriculture and is one of the fields that can benefit from the application of machine learning. Through its heavy research, it is possible to conclude that the most explored algorithms are Neural Networks, but the paper does not provide deeper information about the actual results of these applications.

In [7], the authors present, through the collection of land data using a WSN, a study where machine learning algorithms (SVM and RF) are applied in order to understand the irrigation needs of the land under study, with an accuracy in the order of 80%. Although the study developed in [7] has an intensive research on the level of collected data, and there has been an investment in the analysis of the values through formulas that allow the calculation of the necessary water values, with regard to the algorithms used, the whole Machine Learning solution was based on previous research on the algorithms and the development of the dataset. This may lead to a poor adaptation of the developed technology to the solution where it will be applied.

Our previous research, [8], showed that it is possible to reach savings up to 40% in water consumptions, only through the study of formulas that calculate the amount of water that needs to be administered to the fields. Therefore, the implementation of machine learning algorithms, as this paper will study, is in a good position to achieve even better savings results.

## 3. Materials and Methods

To achieve the proposed goal, the system has a WSN capable of collecting the necessary data and sending it through LoRa and MQTT to the server. This same WSN also has the capability of receiving messages from the server to perform the necessary tasks, such as starting the water pumps during the required time. Besides the messages exchange capacity, the system includes a mobile application that allows consulting the collected data and manage the field remotely.

Therefore, to cope with these requirements the system architecture followed is presented in Figure 1.

### 3.1. System Architecture

Keeping this in mind, it was necessary to create several nodes, each one for a specific task, that together create a WSN, with a star topology, allowing them to share the necessary information among each other. The system needs to include several Sensor Nodes, for data collection, an Actuator Node, to turn *ON/OFF* the irrigation system, and an Aggregation Node, to manage the network and send/receive messages from the server.

Although each node has distinct characteristics, there are certain specifications that are common to all, such as the microcontroller and the communication module that provides the communication within the field network. Therefore, all the nodes share a common base, composed by an ESP32, a dual core microcontroller produced by Espressif [9] and a RFM95W LoRa transceiver.

Regarding the communication module, which ensures the interaction between all the nodes of the network, the LoRa protocol has been used, due to its low energy consumption, long range, low cost of the devices and being a protocol that supports bi-directional communications [10]. The RFM95W module allows long-distance communication with a low consumption relation, being able to transmit up to 2 km away with just 70 mA [11].

Bearing in mind this core system and the needs of the system, each of the developed nodes, their constitution, responsibilities and features are described in the following subsections.

#### 3.1.1. Aggregation Node

The Aggregation Node is the one responsible for exchanging data from the other two types of nodes of the network and subsequently with the server.

As explained above, the core of the node is composed by an ESP32 and a RFM95W LoRa module, to enable the exchange of messages among the nodes in the network, and to communicate with the server MQTT was used.

To use MQTT, an internet connection was required, and, for this purpose, a cellular solution was the best option. In this case, NB-IoT technology was used, since coverage in Portugal is increasing and the technology was developed especially for situations like this, to send small packets of data via a mobile network, allowing small devices to establish internet connection anywhere in the world. To use NB-IoT, the SIM7000E module was attached to the aggregation node, allowing the use of mobile network to establish the Internet connection, with the capacity to work through NB-IoT or 2G technologies. The fact that this module can work using 2G brings advantages as agricultural fields are characterized by low network coverage.

#### 3.1.2. Sensor Node

The Sensor Nodes, as the name indicates, are the nodes where the sensors are connected and responsible for collecting field data. These nodes need to have the necessary sensors to collect the data, a LoRa radio module in order to send the collected data to the Aggregation Node, a control board and a power supply.

The ESP32 microcontroller is used to control all the modules, just like the Aggregation Node. In this node, unlike the Aggregation Node that needs to be always listening, it is possible to use the ESP32 deep sleep functions in order to save energy since the collection of values is timed and the node can be in deep sleep mode during the time of waiting between collections. To send the collected data to the Aggregation Node via LoRa a RFM95W radio is used as explained before.

Regarding the data collection, several sensors were used, such as:SI7021, a temperature & humidity sensor, capable of collect temperature and moisture values with high precision (±3%RH, ±0.4 ${}^{{}^{\circ}}$C), in a range between −40 ${}^{{}^{\circ}}$C and 125 ${}^{{}^{\circ}}$C in terms of temperature and 0 to 100% in terms of humidity values [12].DS18B20, a waterproof soil temperature sensor, capable of collect data between −55 ${}^{{}^{\circ}}$C and 125 ${}^{{}^{\circ}}$C with an accuracy of 0.5 ${}^{{}^{\circ}}$C for −10 ${}^{{}^{\circ}}$C to 85 ${}^{{}^{\circ}}$C interval, easy to implement, with a one wire connection [13].Analog Capacity Soil Moisture, a humidity waterproof sensor which provides the capacity of collecting data with high precision. This collection is made by capacitive sensing [14].

To keep the data of the field under analysis updated, a collection is made every 10 min. Between each collection, and in order to save the battery that powers the node, the circuit is placed in Deep Sleep mode where the energy consumption is reduced to 100 μA. At the end of each Deep Sleep time the node wakes up and runs all the code again, collecting and sending new data.

To power the nodes, due to its low power consumption, it becomes possible to use batteries. To make the system more complete and reduce the need for maintenance, a solar panel was also used to charge the battery, ensuring that they will always have enough energy to power the node even during long periods of absence of sun exposure.

Through the analysis of the hardware datasheets it is possible to assess that the electrical current of the node, in his two states, is 0.1 mA in Deep Sleep mode and 92.18 mA in active mode, with a 604 s cycle, collecting every 10 min (600 s), taking 4 s to collect and send the data.

Regarding the calculation to find the correct size of the solar panel to use, Lisbon gets 19.6 $W/m^{2}$ of solar radiation [15], already accounting with the low efficiency of solar panels, that achieve only about 15% of the total solar radiation, and the efficiency of the charging module and the charging process itself, with only about 10% of the solar irradiance being used at the end of the process [16].

To calculate the needed panel size, Equation (Equation 1) was used, where *S* is the size of the panel in m${}^{2}$, *U* is the electrical potential in Volts, *I* the electrical current in Amperes and *A* the average electrical power by area unit in Watts per square meters (W/m${}^{2}$).
(1)$$S=\frac{U\times I}{A}$$

Since the nodes work at 3.3 V and with a current of 0.1 mA and 92.18 mA, depending on the cycle stage, the results given by the presented equation shows that the solar panel area for Deep Sleep mode is minimal, 0.153 ${\mathrm{cm}}^{2}$, and the area for active mode is 155 ${\mathrm{cm}}^{2}$. Considering that every cycle last 604 s and only 4 s is related to active mode, about 0.7%, and with the aim of creating some margin, it was considered 20% of the calculated area for active mode, 31 ${\mathrm{cm}}^{2}$.

The efficiency of a solar panel is mostly influenced by the daily solar exposure, being this variable through the year, since in the winter days the solar exposure decreases in a large scale. To solve this situation and ensure the power supply of the node during the entire year, it was studied the duration of the used batteries. A 3.7 V LiPo battery with 4000 mAh (14,400,000 mAs) was used in the system, given the system consumption in both modes and the battery capacity, was then possible to calculate the number of cycles that the battery supports, through Equation (Equation 2).
(2)$$N_{\mathrm{Cycles}}=\frac{{\mathrm{Cap}}_{\mathrm{battery}}}{C_{\mathrm{DeepSleep}}+C_{\mathrm{Active}}}=\frac{14400000\;\mathrm{mAs}}{0.1\;\mathrm{mA}\times 600\;s+92.18\;\mathrm{mA}\times 4\;s}=33645$$

Considering that each cycle lasts 604 s, meaning that 5.96 cycles are done per hour, through Equation (Equation 3), it is possible to calculate the duration of the chosen battery in days.
(3)$$N_{\mathrm{Days}}=\frac{N_{\mathrm{Cycles}}}{5.96\times 24\;h}=235\;\mathrm{days}$$

It was then possible to understand that through the use of a solar panel with about 31 ${\mathrm{cm}}^{2}$ of area and a battery of 4000 mAh, it was possible to ensure the correct power supply of the developed sensor nodes, ensuring that the supply system could give the needed power to supply the node over 235 days of no solar exposure.

#### 3.1.3. Actuator Node

The Actuator Node was in charge of acting according to the collected data by the Sensor Node. This node consists of a microcontroller, ESP32, a LoRa radio module, RFM95W, the weather station and the irrigation pumps. As explained before the RFM95W will be responsible for exchanging messages between the node and the Aggregation Node, receiving the irrigation instructions and sending the current status of each pump.

The microcontroller will receive the update messages for irrigation and process them to control the pumps during the needed time for each section. It will also be connected to a weather station, the SEN 0186 [17], capable of collecting measured wind speed, wind direction and precipitation values. These sensors are not connected to a Sensor Node, since they need to be constantly collecting data, not being able to work with batteries.

Regarding the node power consumption, and since the irrigation pumps used work at 12 V, it was not possible to power the node by batteries or solar panels, similar to the Sensor Node. Therefore, the node will have to be powered by an electrical socket with a transformer that converts the 220 V of the local electric current to 12 V needed. Also, a current converter, the LM2596, was used to convert the 12 V coming from the transformer to 5 V, in order to power the ESP32 and the whole circuit.

Since ESP32 does not have enough power to supply the water pumps the Panasonic AQY212EHAT Solid State Relays (SSR) was used, making possible, through a simple electronic circuit and HIGH and LOW signals from ESP32, to switch *ON* or *OFF* the 12 V supply to the water pumps, switching there status.

### 3.2. Data Analysis

As the main goal of this work is to create an autonomous way to improve the efficiency and sustainability of irrigation in agricultural sites, based on an IoT and WSN system, a way to analyze the sensor data collected and transform those information into knowledge is needed, mainly to control the real amount of water needed or the best time of day to irrigate. For that, two different approaches were done, one only based on calculations using the sensor data, to discover the optimal irrigation time, and a second one based on a machine learning approach, using the sensor data to predict the best irrigation hour and then calculate the irrigation time, with the previous approach.

#### 3.2.1. Irrigation Algorithm

Taking into account the main objective of the developed system, water saving formulas were studied and implemented in order to calculate how much water is necessary to administer to a specific irrigating zone based on the data collected by the sensors on that zone.

These formulas were developed by our previous studies [5,8] and were developed “taking into account the use of soil moisture and air temperature and humidity sensors and considering the type of crops, the type of valves and tubing used, the distance between these same valves and the number of irrigation in one day. Considering all these parameters and using Equation (Equation 4), it becomes possible to calculate the amount of water for each irrigating zone.
(4)$$T=\frac{A\times (K_{c}+\mathrm{ET})\times 60}{F\times N\times 1000}$$

In Equation (Equation 4), the result, T, represent the irrigation time in minutes, $K_{c}$ is the crop coefficient, according to [18], ET is the evapotranspiration in mm/day, resulting from Equation (Equation 6), F is the outgoing flow of water in $m^{3}/h$, N is the number of valves, P is the number of irrigation periods and A is the garden area in $m^{2}$, this last is given by Equation (Equation 5).
(5)$$A=\left[\left(0.5\times N\right)-1\right]\times D^{{}_{2}}$$

In Equation (Equation 5), *D* represents the distance between valves.
(6)$$\mathrm{ET}=0.0023\times \left(T_{\mathrm{med}}+17.78\right)R_{o}\times {\left(T_{\max}-T_{\min}\right)}^{0.5}$$

In Equation (Equation 6) it is possible to see the simplified Hargreaves formula [19]. This one will give the evapotranspiration (ET) in mm/day, where $T_{\mathrm{med}}$ is the average temperature, $T_{\max}$ the maximum temperature, $T_{\min}$ is the minimum temperature and $R_{0}$ is the incident extra-terrestrial solar radiation in mm/day, according to [20].

To account for the sensor values from soil moisture and air humidity, in order to get an optimized time for irrigation it is possible to formulate Equation (Equation 7).
(7)$$T_{\mathrm{opt}}={\mathrm{TI}}_{\mathrm{soil}}\times 0.7+{\mathrm{TI}}_{\mathrm{hum}}\times 0.3+T\times 0.1$$

In this, $TI_{x}$ is the time needed to reach 100% of the variable x, soil moisture (soil) and air humidity (hum), respectively, given by Equation (Equation 8), where $I_{x}$ is the last value collected for sensor x.”
(8)$${\mathrm{TI}}_{x}=\frac{T\times (100-I_{x})}{100}$$

Besides this approach gave a 34% improvement when used in a grass irrigation system in [8], as we stated, they can be still improved using more sensor data to optimize even further the formula. For that, and since our proposal adds a weather station to the system, we can improve the formula using the values for rain and wind.

As such, the first thing to add is the effect of rain in irrigation. Equation (Equation 4) shows that for irrigation the crop coefficient, $K_{c}$, summed with the evapotranspiration value, ET, represents the mm/day of water needed for a healthy crop. However, when rain falls in that area that amount of water might already been put into the crops, and using the rain gauge from the weather station it is possible to know exactly the amount of rain in the last 24 h, so that value can be subtracted from the previous one, indicating exactly the amount of water the crops still need for that day. So it is possible to add the last amount of rain for the last 24 h, $TI_{24\mathrm{rain}}$, to Equation (Equation 4) and optimize that formula.
(9)$$T=\frac{A\times (K_{c}+\mathrm{ET}-{\mathrm{TI}}_{24\mathrm{rain}})\times 60}{F\times N\times 1000}$$

Finally, and as the system must react to real time changes in the environment, the following methodology was used to stop the irrigation due to rain or strong winds, as if it is raining there is no need to be irrigating and if strong winds occurs, they can blow away the water, so irrigation must stop and be scheduled for another time. For that the threshold of 1.0 mm of rain in the last 5 min or wind speeds over 35 m/s were set to indicate whether irrigation can start or if it is scheduled for another time.

#### 3.2.2. Machine Learning

As explained in the previous topic, it became possible to estimate the time of irrigation needed to be administered to an irrigation zone, however the time of day when this water administration was done had to be indicated manually by the user, leading to irrigation being administered at a less favorable time of day, leading to water wastage. To make the system more complete, by removing the need for manual irrigation time input, a machine learning algorithm was developed to predict the most appropriate time of day for water administration.

The developed script implements a machine learning algorithm that after receiving as input the current time, air temperature, air humidity, wind speed and direction, soil humidity and whether or not it is favorable for watering, the script returns the most suitable time to irrigate the zone under study.

The methodology for this approach, as shown in Figure 2, is to receive the data in real time from the system, pre-process them and put them through the machine learning algorithm in order to understand, based on a previous trained model and dataset, if that hour is suitable for irrigation and if not, which will be the best hour. From time to time, the model is retrained using a new dataset that includes all the new data that was collected since the last train.

The study of the best model, the dataset creation and the data pre-processing will be explained in detail in Section 4.

### 3.3. Data Visualization

To analyze the collected data and perform the needed tasks, a mobile application was developed, to offer the user the possibility to perform tasks without the need to be in the field location.

The developed application has the functionality of showing the collected data and the possibility to perform tasks. For this, it was necessary to give it the ability to connect to the developed API and the tools to use MQTT in order to perform tasks into the field, for example turning ON one actuator.

To give the right logic to the application it was necessary to create multiple views, Figure 3, being each panel shown in the application and responsible for specific features.

After the authentication process, the dashboard presents to the user all the systems that he has connected to his account, each one representing a different agricultural field, Figure 3c). Here the user can select an individual field and is forward to a new screen showing all the sections associated with that agricultural field, Figure 3d), each representing an individual irrigation zone.

Just like the fields view, by choosing and clicking in one of the irrigation zones, the application leads the user to the data view, Figure 3e), where the last collected values from the sensors installed in the selected zone are shown. To analyze the variation of the collected values from each sensor, by clicking on the desired sensor, a new view, Figure 3f), presents a chart with all the collected values from that sensor and the corresponding timestamp. The final view, Figure 3g), allows the user the possibility to perform tasks on the field, mainly in order to change the irrigation zone settings, consult the local weather forecast, turn *ON* or *OFF* the water pump of each irrigation zone and turn *ON* or *OFF* the auto irrigation mode.

## 4. Data Analysis Methodology

With the constant evolution of technology and the appearance of new solutions that, when combined, manage to achieve sustainability, the exploration of these systems is increasingly a path to take. This way a study and development of machine learning algorithms was made with the aim of predicting the most suitable time of day for water administration to an agricultural field.

Machine Learning (ML) is a technology which has the capacity to learn and improve through his own experience. These improved capacities are possible due to the access of a huge amount of data previously granted to the system and which is constantly updated with new data, as the algorithm is exposed to new situations and must give an answer for that. One of the great characteristics of ML is that all this constant and automatic learning is done without any human interaction.

To study the most suitable classification algorithm, the following machine learning algorithms were considered: Random Forest (RF), Neural Networks (NN), Decision Trees (DT) and Support Vector Machine (SVM). In addition, and in order to be able to study correctly the algorithms already mentioned, a methodology was created, leading to the best possible optimization.

### 4.1. Machine Learning Classification Models

To achieve the pretended ML algorithm, there are some learning techniques that can be used, including supervised and unsupervised learning. Supervised learning is one of the most used techniques to develop ML algorithms, this technique is based on providing samples (entries) in a dataset where, during the training process, is provided to the developing algorithm both the input as the output (the one that the final algorithm as to predict after the training phase). Each entry of the dataset is characterized by an established number of features, being the same for all entries. Regarding unsupervised learning, this technique is known as the process of the training without the knowledge of the complete dataset, being the training process focused on data discovers and the find of hidden patterns [21].

The supervised learning is divided in classification and regression. Classification methods are known for their goal of approximating a mapping function from the inputs given by the dataset, in order to identify the output values. In Equation (Equation 10) is shown the followed function by this method, where f is the mapping function, X the input value and Y the output value [22].
(10)y(f:x→y)

#### 4.1.1. Random Forest

Random Forest (RF) is a tree-based method that conglomerates several self-determining decision trees developed for classification and regression. Through the combination of the various trees it is able to understand which is the best option, being the main objective to reach one in pure i.e., a node formed by a single class, giving it high predictive capabilities [23].

#### 4.1.2. Decision Trees

Decision Trees (DT) are tree-based methods in which each path begins in a root node representing a sequence of data divisions until reaching a Boolean outcome at a leaf node. These methods can be applied for classification and regression. The final goal of this method is to reach a model that can predict the search value for that specific scenario by learning simple decision rules [24].

#### 4.1.3. Support Vector Machines

Support Vector Machines (SVM) are a set of supervised learning methods developed for classification, regression and outlier detection which is known by his high effective in high dimension spaces and for its use for training points in the decision function, being also memory efficient [25].

#### 4.1.4. Neural Networks

For the study of Neural Networks algorithms, which are defined as computational models of nervous system, the Multi-layer Perceptron (MLP) method was used, which is a supervised learning method that learns a function $f\left(.\right):R^{m}\to R^{o}$ by training on a data set, where *m* is the number of dimensions to input and *o* the number of dimensions for output. These MLP networks are characterized by being general-purpose, flexible and non-linear. Their complexity can be changed according to their application by varying the number of layers and units of each layer [26,27].

### 4.2. Methodology

To train the classification model, the following steps were done, using Python, the scikit-learn libraries [28] and the Anaconda environment.

For each algorithm a model was trained using the corresponding dataset and the default configuration parameters. This allowed for quick comparison of the performance, in terms of accuracy, of each model and understanding which are more likely to guarantee best results and which need to be improved to achieve them. The scikit-learn, an open source Machine Learning library developed for Python implementation [28], was the selected framework for the development of the machine learning models.The obtained model for each algorithm is submitted into a hyper parametrization tuning, that compares the model performance using different model configurations parameters, to understand which is the configuration that obtains the best performance, facing the dataset and the goal. For this, a method provided by scikit-learn called RandomizedSearchCV was used, which performs the fit and training of the algorithm under study, calculating which parameters are best suited to it [29];To guarantee that the model is stable, after finding the best configuration and training the model, a Stratified K-Fold cross validation is performed, in order to guarantee that the model is not under or over fitted. Using five folds, it is possible to use a different set of training and validation data on each fold, allowing for the model to check on every single datapoint. This way, it is possible to really understand the model performance, as each of the folds will produce a result, that is averaged at the end, allowing for a reduced error margin and variation, as more data is used to fit the model;

After the cross-validation, and to be able to compare between the models trained with the same dataset, a statistical hypothesis-testing was performed using a paired statistical test and the 5×2 fold approach, assuming a significance threshold of α=0.05 for rejecting the null hypothesis that the algorithms perform equally well on the same dataset.

### 4.3. Dataset & Data Pre-Processing

To develop the correct algorithm that could predict the best hour of day for water administration, it was necessary to compose a correct and useful dataset.

With the aim of collecting the necessary amount of data needed to create the pretended dataset, a prototype of our system was used to collect the necessary data, for air temperature and humidity, soil temperature and moisture, rain and wind speed and direction, in a small urban farm. For that, three sensor nodes and one actuator node, including the gateway and weather station, were installed in three different zones of an urban farm during 12 months, between August 2019 and August 2020, to collect the needed field data to train the model. The nodes collected information every hour about each zone and weather conditions as well as the irrigation periods, timing and method. This allows us to collect 26,280 data points for field conditions and irrigation needs. The selected sensors were chosen based on its importance upon our goal, not only the needed inputs for the presented irrigation algorithm in Section 3.2.1, but also critical information about the field conditions that can help the developed model to understand the field needs.

After the conclusion of the collection phase, the data was compared and complemented, when necessary, with values provided by IPMA, the Portuguese Weather Institute, being able to confirm the accuracy of some of the collected data since some of these were available on IPMA API [30]. This API provides hourly data from all the weather stations in Portugal, containing historical data on air temperature, air humidity, precipitation, wind speed, wind direction, solar radiation, evapotranspiration and more. The data collected from the API was used to compare to the sensor data and understand if the sensors were able to gather precise information about the environment and weather conditions and also to assess if the weather sensors could be complemented or exchanged with only API data from weather stations. Table 1 shows the variance between sensor data and API data for the weather conditions.

It is possible to check that some variations exist between the sensor data collected directly in the field and the ones collected by the Weather station 1 km away. This can be justified by the distance between the collection points but also for the precision of the used sensors on our system, for example for the temperature and humidity sensors that are ±0.4 ${}^{{}^{\circ}}$C and ±3.0%, respectively. It is possible to assess that the sensors performed as good as the weather stations and as such are always a valuable addition to the system, being able to collect data directly in the field and not depending on third parties weather data that can be collected on points further away from the system implementation.

To complement our dataset the closest weather station was selected, being this around 1 km away from the urban farm. Since the API provided hourly data starting from 2014, the field data collected for 2019 and 2020 was used to estimate the field data since 2014, in order to have a larger dataset.

Besides the features provided by the implementation of the presented system and API information from IPMA, all entries were individually pre-processed and analyzed, with extra features being added to each one, making the dataset richer and more substantiated in order to improve the training process and achieve better results. These new features are: “Is_Favorable”, “Need_Irrigation”, “Had_Irrigation” and “Suggested_Hour”.

Regarding the “Is_Favorable” feature, this one refers to the weather and field conditions, indicating whether they point to favorable or unfavorable conditions for sustainable irrigation. This feature was calculated through some conditions, such as: the wind speed under 36 m/s; the wind speed under 25 m/s and the wind direction between 180 and 359 degrees; the rainfall values under 1 mm/h; the air temperature under 30 ${}^{{}^{\circ}}$C; the hours not in between 10:00 and 16:00. If all these conditions were verified, it was considered favorable for irrigation.

The “Need_Irrigation” feature, was calculated through two conditions, as the feature presented above, being these as follow: the soil moisture under 60%RH with the air temperature over 25 ${}^{{}^{\circ}}$C or the air humidity under 40%RH; or the soil moisture under 45%RH, disregarding weather conditions. If one of these conditions is checked, the field needs to irrigate.

For the “Had_Irrigation” feature, it only indicates if the field was irrigated in the day in question.

Finally, regarding the “Suggested_Hour” and since this would be the features that the Machine Learning had to predict, this feature was added manually to each entry, ensuring that the suggested hour was adapted to each weather and field condition.

After conclusion, the dataset used has 105,217 entries and had the following parameters:Year—Year of the observationMonth—Month of the observationDay—Day of the observationHour—Hour of the observationTemperature—Air Temperature registered [${}^{{}^{\circ}}$C]Relative_Humidity—Air Humidity registered [%]Total_Precipitation_Low—Precipitation registered [mm/day]Wind_Speed—Wind Speed registered [km/h]Wind_Direction—Wind Direction registered [${}^{{}^{\circ}}$]Soil_Humidity—Soil Moisture registered [%]Had_irrigation—Field irrigated [0/1]Need_Irrigation—Field needs irrigation [0/1]Is_Favorable—Conditions favorable for irrigation [0/1]Suggested_Hour—Suggested irrigation hour [0–23]

Table 2 provides some statistical information regarding the chosen dataset and each of its features.

Before the study of the chosen algorithms, a test was performed in order to understand which were the most important parameters of the dataset. The results obtained can be seen in Figure 4.

As can be seen, the features “Day”, “Need_Irrigation”, “Month”, “Had_Irrigation” and “Total_Precipitation_Low” have a low importance for the training and later result of the algorithms, so they can be discarded. This discarding also results in a shorter time in the training of the algorithms and no high variation was observed in the results obtained after the discarding.

### 4.4. Model Analysis & Remarks

The presented training methodology was followed in order to obtain the best model possible to predict the best irrigation hour, based on field conditions and weather forecast.

To train, validate and test the model, the presented dataset will be used, being divided into three groups: 70% for training, 20% for validation and 10% for testing. To evaluate the model performance, and since a classification is used, the Accuracy metric will be used, as it is the most common metric for classification. It measures the fraction of predictions the model got right, using Equation (Equation 11).
(11)$$\mathrm{Accuracy}\;\left[\%\right]=\frac{\mathrm{Number}\;\mathrm{of}\;\mathrm{correct}\;\mathrm{predictions}}{\mathrm{Total}\;\mathrm{predictions}}$$

The estimated data nearly matched the real data when Accuracy is near 100%.

The results obtained from the training methodology can be found in Figure 5.

The first thing to notice, in almost every model, is that the hyper parametrization values always have a higher accuracy, followed by the cross-validation values and finally the default values. This is justified by the methodology followed, as the default model in the first step is tuned to improve the default result, obtaining always a better result. Then, in the cross-validation step, as multiple combinations of the dataset are tested, and the accuracy for each fold is averaged, it is expected that the accuracy decreases. Although this happens, the cross-validation results allow for a better knowledge of the model accuracy, as they were exposed to a higher variety of unknown data.

Considering only the cross-validation results, as they are the ones best fitted to evaluate the model accuracy, Figure 6 shows the standard deviation of each of the models after the cross-validation.

The best classification model for predicting the best irrigation hour, based on environmental and field sensor data, is Random Forest, with an accuracy of 84.6%. Neural Networks were the second best model, with 80.2%, followed by SVM and Decision Trees, with 79.3% and 77.0%, respectively.

As such, although with only 85% in accuracy, a value that can still be improved, Random Forest is the best classification model and it will be the one used in the implementation scenario.

To further validate this conclusion, as described in the methodology, a paired statistical test using the 5×2 fold approach, comparing the Random Forest model against each of the other models. Table 3 shows the obtained *p*-values.

Since all *p*-values are lower than the defined α of 0.05, it is possible to reject the null-hypothesis that the other models perform equally well as Random Forest on this dataset, meaning that algorithms are significantly different, and concluding with more certainty that Random Forest is the best classification model for our methodology.

Compared to the results found in the literature, where [7] achieved an accuracy of 80% using Random Forest and [31], that used a similar dataset, also having results in the 84% accuracy for Random Forest, it is possible to conclude that Random Forest is indeed the best algorithm to predict irrigation scheduling. Other solution can be found using other models, such as Dynamic Neural Networks [32], that achieved a 10% margin of error for irrigation scheduling based on soil moisture sensors, and [33], that schedule irrigation based on soil moisture predictions using XGBoost, with a 13% margin of error. In this last study, it is also possible to check a study done on the same algorithms as our study, and between them, Random Forest was also the best solution, with a margin of error of 14%, similar to our 85% accuracy.

Our research improves upon these results, as it uses a more complete dataset, including more sensor data that can be collected in the field, not relying only on soil moisture and using data that affect water dispersion and evaporation, such as temperature and wind, allowing for a more detailed evaluation on the best irrigation hour.

## 5. Implementation

To understand how the system performs when exposed to the real environment, the system prototype was installed in a small urban farm.

Field management was performed by the owner, who watered it once a day, around 20:00, using a hose, being the only days that it was not irrigated, the rainy ones. Figure 7 shows the urban farm used for the test.

The managed field was divided in two irrigation zones, with each one assigned the following sensors: air temperature and humidity, and soil moisture and temperature sensors. Besides that, the data of wind speed, wind direction and the rainfall were also collected by the weather station. The sensors implementation on the field can be seen in Figure 8.

Regarding irrigation, it was installed one Actuator Node that managed two water pumps, each one attached to each irrigation zone, as seen in Figure 9, including also the weather station, as demonstrated in Figure 10.

In the first stage, and since the proposed system has a high component of data collection, it was performed in a data collection test in a real environment. The goal of this test was to ensure the correct data collection by the system, so that when the actuators were installed, the collection part works correctly, and to understand how the different irrigation zones react when exposed to the same situations. To guarantee the correct outcome is seen, this test was performed for three weeks.

After the collecting test and knowing the system architecture is capable of gathering and transmit the data information, the full system was tested. To inquire the developed methodology, in zone one was implemented the Watering Algorithm, which verifies every hour if there are some zones to be irrigated, in case of matching, the script calculates the amount of water needed for that zone. Since the owner usually irrigates his garden once a day, with the exception of rainy days, at the same hour, it was programmed that the irrigation zone that this algorithm was taking care of would be irrigated at 20:00, every day, according to the algorithm decisions.

In zone two, the Watering Algorithm and the developed Machine Learning algorithm were implemented. By implementing these two algorithms in parallel, the field was autonomous, since neither the irrigation hour nor the irrigation time had to be entered manually into the system.

To expose the system to different situations, the second stage of the test was in operation for three months, between September and November of 2020, where it was exposed to good weather and rainy days.

The main goal of the presented test was to estimate the amount of water that can be saved through the implementation of the developed solutions, always ensuring the quality of the field conditions, and also compare to the usual amount of water used manually by the user.

## 6. Results

This section presents the obtained results for the two stage implementation described previously, starting with the data collection test, that verifies the system architecture capabilities of gathering and transmitting the field conditions, as well as the differences between the two irrigation zones, and then the full system implementation, with the intelligence and autonomous irrigation modules, in order to understand if a more sustainable and efficient irrigation system can be achieved.

### 6.1. Data Collection Test

The results obtained from the test showed a variety of results from which some conclusions could be drawn. Since the performed test collected values of air and soil temperature, it was possible to observe the differences between these two. In Figure 11, it was possible to observe the air and soil temperature variation over one day.

In Figure 12 it is possible to observe the rainfall values variation during a rainy morning and the moisture values collected for the same day by the two soil moisture sensors installed.

The collected values will allow for, not only, understand if the system architectures work as intended, but also understand how these parameters affect the field conditions and needs.

### 6.2. System Performance

After testing the system capability of gathering and transmitting data, it was possible to implement the full system to autonomously control the irrigation system.

To evaluate the improvements, in a first moment, the amount of water spent by the user in each irrigation zone was analyzed and it was concluded that the hose has a water flow of 23 L of water per minute and, to water each irrigation zone, the user takes 25 s to do it. Therefore, through the calculation 23 l/min×25 s=5.75 l, it is possible to understand that 5.75 L of water are used to irrigate each irrigation zone per day.

In a second moment an analysis was made on three distinct situations to which the system was exposed, such as: sunny days, rainy days and days after rainy days. This allows us to understand how the system and each approach react to different situations. For each analysis 3 days were taken into account. For each of these days Table 4 presents the time irrigated and the water used and is followed by a detailed results description for each scenario. The presented results were obtained based on the water flow of the used water pumps, which was of 5.83 L/min.

Figure 13 shows the system behavior for the sunny days, where is it possible to observe the Soil Moisture values for both zones and the moments that irrigation took place, marked for each of the developed algorithms, with the Watering Algorithm tested in irrigation zone number one, identified by 1, and the combination of the Watering Algorithm and Machine Learning Algorithm tested in irrigation zone number two, identified by 2.

In a second moment, the behavior of the system for rainy days was analyzed, with Figure 14 presenting the soil moisture values variations of the two irrigation zones under test and the rainfall values.

In a third and last moment, the behavior of the two implemented methods for days after rainy days is presented in Figure 15.

Table 5 presents the results for the entire test, for the three months, based on water used to irrigate the field in each exposed situation, where the third method is the one used by the owner to irrigate his garden.

## 7. Discussion

In this section, the previously presented results for both stages of testing will be discussed, and conclusion will be taken.

### 7.1. Data Collection

The collected data showed that, in terms of air and soil temperature, represented in Figure 11, they both follow the same variation with air temperature influencing the soil temperature. Although it is possible to check that when air temperature changes abruptly, soil temperature takes a more gradual variation, being 4 ${}^{{}^{\circ}}$C lower, on average, than air temperature, and achieving their maximum values an hour later, on average, and minimum values, only 25 min after the air temperature.

Regarding the difference between the collected data from the two irrigation zones, although the distance between them is only one meter apart and the weather conditions being the same, it is possible to observe a difference in the variation of their values, where it is possible to conclude that irrigation zone number two does not reach such high temperature values when compared to zone number one. This will have an impact on the need for irrigation, contributing to different needs for the same crop in different zones.

Another analysis which was possible to make, through the collected data, was the relation between the rainfall and the moisture values in the two irrigation zones. Through the analysis of Figure 12 it is possible to check that 32.26 mm of rainfall for four hours, and since it starts it impacts the soil moisture values. Through the analysis of the graphs, it is possible to observe the increase of soil moisture values from about 30 min after it started raining. However, the collected values of each zone were quite different, showing that the soil moisture values of zone one were always lower than in zone two and, when started to rain, the values of zone two increased in an abrupt way and, in contrast, the values of zone one have gradually increased. After the rain stops, both zones converge to a similar value of soil moisture, with zone two losing soil moisture whereas in zone one it was slightly increasing during time. This shows that zone two absorbs water more easily and zone one absorbs water over time.

With the collected values and comparison between the two tested zones it is possible to establish the need for individual configuration of the zones, as they react differently in terms of temperature and moisture, meaning that irrigation times can be different for each zone. This proves that our methodology for an autonomous system that adjusts the irrigation based on the collected sensor data is more efficient than using the same time for all zones.

### 7.2. System Performance

Considering the three scenarios under test, sunny days, rainy days and days after rain, for both zones using different approaches, only the Watering Algorithm and the combination of the last with the Machine Learning algorithm, some conclusions can be taken.

For the sunny days, through the analysis of Figure 13 and Table 4 it is possible to notice that during the three analyzed days, the Machine Learning algorithm only pointed out the need for water administration for one of the days, also indicating that the best hour of the day to administrate that water was 20:00, with no need for water administration for the other two days, using 52 s of irrigation and 5.05 l of water. The Watering Algorithm irrigated every day with a combined time of 80 s (32, 24, 24, respectively for each day) and 7.77 l of water used. This shows that on a sunny day the system using the machine learning approach saves 53% more water that when using the watering approach.

In the rainy days, represented in Figure 14, it is possible to notice that neither the Machine Learning nor the watering algorithms indicate the need for water administration for any of the presented moments, since, as can be seen in the collected values, these days had long periods of rain which led to an increase in soil moisture values and trigger the rain threshold, that disables irrigation if 1 mm of rain is registered in the previous 5 min.

As for the days after rainy days, through the analysis of Figure 15, it is possible to notice that the Machine Learning algorithm only indicated the need for water administration on the third day without rain, also indicating that the best time for it to be administered was at 20:00, with 49 s of irrigation and 4.76 l of water used. The watering algorithm irrigated each day for a combined time of 53 s and 5.15 l of water. This shows that the machine learning algorithm, in a day after raining, saves 8% more water than the watering algorithm.

As can be observed, the irrigation times for each situation depend on what the system was exposed to, allowing us to understand that the correct use of water varies from day to day and is important to make those changes each day to create a more sustainable irrigation process.

Through the analysis of Table 5, is possible to understand that the method which had a least water consumption was the one that implemented both developed algorithms, using machine learning and the watering solution, understanding the water need for the three tested situations and adjusting the amount of water administered for each scenario.

Regarding the first method, it is possible to understand that without the hour indication for watering, this method does not achieve the best possible results since it cannot understand which were the weather conditions, needing an hour indication to perform irrigation every day. However, excluding the rainy days, the water values calculated by the watering solution, were lower than the used ones by the owner with his daily manual irrigation, by 50.26%, saving more than 179 l of water in 90 days.

It is then possible to conclude that, through the use of the combination of the two developed algorithms, was possible to save around 60% of water when compared to water used by the owner initially, saving more than 214 l of water in 90 days, and 19.57% when compared to the other method.

Besides the achieved water saving values, it was also possible to ensure that the system, in any moment, allowed the soil moisture values to reach the minimum recommended values, being this 40%, taking into account that the area where the system was tested is characterized by having high humidity [34].

In the literature and industry, it is possible to find several approaches to smart irrigation using IoT and Machine Learning in order to reduce the amount of water used or calculate the best irrigation period.

In [35], soil moisture sensors and evapotranspiration prediction were used to create a better irrigation scheduling, achieving a 39% reduction in water consumption. In [36] presents a solution capable of reducing 23% of water in irrigation using LoRa nodes that gather information about air temperature and humidity and soil moisture, that were analysed using fog computing. The Smart&Green framework, presented in [33], recommends the optimal irrigation management-based field configuration and soil moisture data, using computational models to reduce the water used in irrigation in 56%. Finally, [32] used Dynamic Neural Networks to predict the soil moisture in the fields and adjust the irrigation process accordingly, achieving up to 46% in water savings.

All these works present a similar methodology to the one presented in this paper, but none achieved the 60% water savings. This proves that combining real-time sensor data from several sensors and machine learning is a better methodology than predicting the data or using only soil moisture.

Our previous work can also be compared, as the same methodology for smart irrigation was used without the machine learning approach [8], and only achieved 34% in water savings.

It is also worth mentioning that in the industry there are already a set of solutions available and ready to use, with almost all using smart sensors and controllers. These solutions, not only are more costly, but in almost every supplier only 50% of savings are provided.

## 8. Conclusions

This paper presents a methodology for a sustainable irrigation system for farming that uses machine learning and real-time data to achieve the best watering strategy. The methodology for the system was presented, as well as the training and validation of the machine learning model, including a comparison between various classification techniques. A practical implementation of the methodology is also presented, not only to validate the methodology, but also to compare the efficiency facing traditional and other intelligent models. For all these scenarios the results were presented.

The first thing to conclude is that with proper adjustments, it is possible to create a more efficient irrigation system using wireless sensor networks, and that our proposed architecture works well in this environment.

Regarding the machine learning model, several classification techniques, including Decision Trees, Random Forest, Neural Networks and Support vector Machines were tested, to assess which performs better in terms of predicting the best hour for irrigation, based on the conditions of the field and weather forecast given by the used sensors. It was possible to conclude that Random Forest was the best solution, achieving an accuracy of 84.6%. Random Forest results surpass SVM and Decision Trees by almost 7%, being Neural Network the only technique that provides results more similar to Random Forest, being only 4%.

Finally, two modes of selecting the irrigation hour and time were presented, each with a specific method. The Watering mode, which follows our previous work presented in [8] with results of about 40% in savings based only on formulas, was improved and can now achieve about 50% of savings, when compared with traditional methods, in this case manual irrigation. When combining the watering method with a machine learning approach, that predicts the best irrigation hour, it is possible to save up to 60% more water than traditional methods, with more than 214 L of water saved in only 90 days.

It is then possible to conclude that the developed system achieved the proposed goals, since it was possible to reach sustainability through an IoT solution using WSN and Machine Learning technologies, presenting a low maintenance requirement system with a low complexity level and high operating capacity when exposed to various weather conditions, and even give control to the user over a mobile application with all the necessary features and requirements to be adapted in order to be introduced in the market.

## Figures and Tables

**Figure 1 sensors-21-03079-f001:**
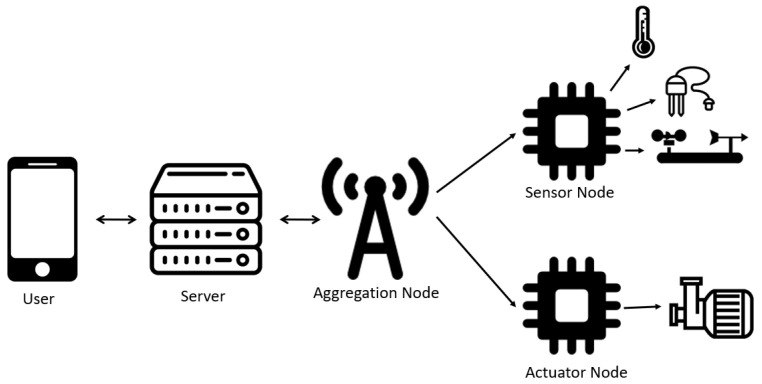
System Architecture.

**Figure 2 sensors-21-03079-f002:**
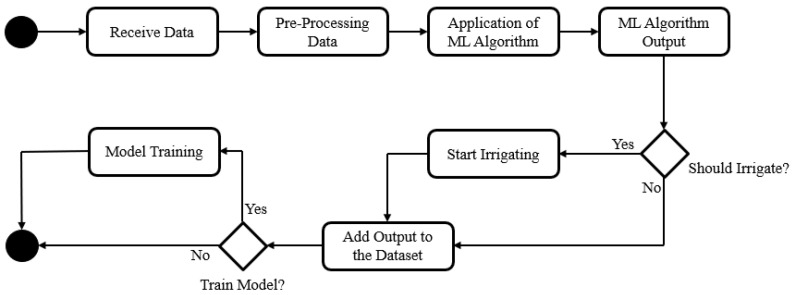
Machine Learning Methodology.

**Figure 3 sensors-21-03079-f003:**
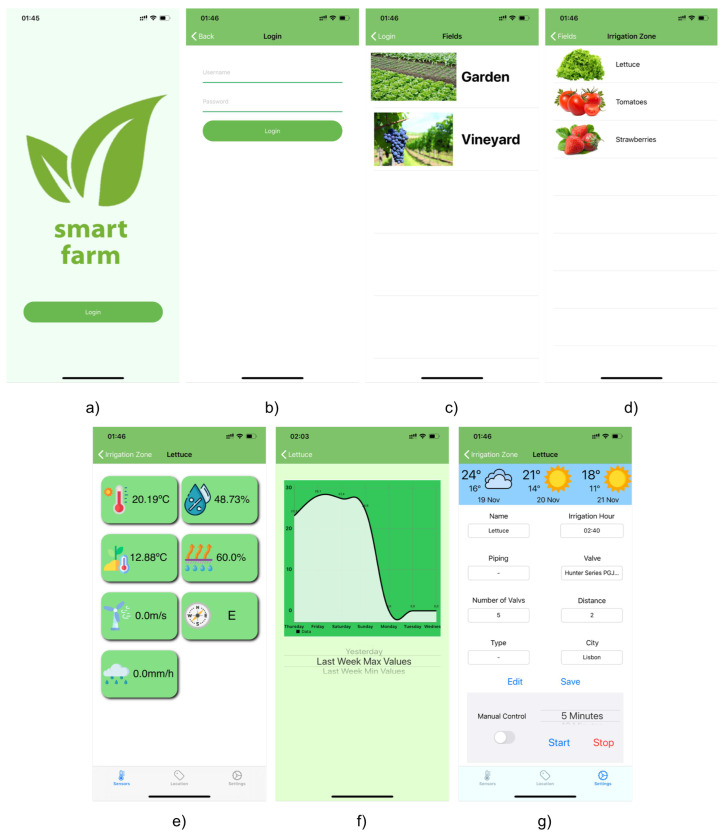
Dashboard Application views—(**a**) Start View; (**b**) Login View; (**c**) Fields View; (**d**) Irrigation Zones View; (**e**) Sensors Values View; (**f**) Sensor History View; (**g**) Irrigation Configuration View.

**Figure 4 sensors-21-03079-f004:**
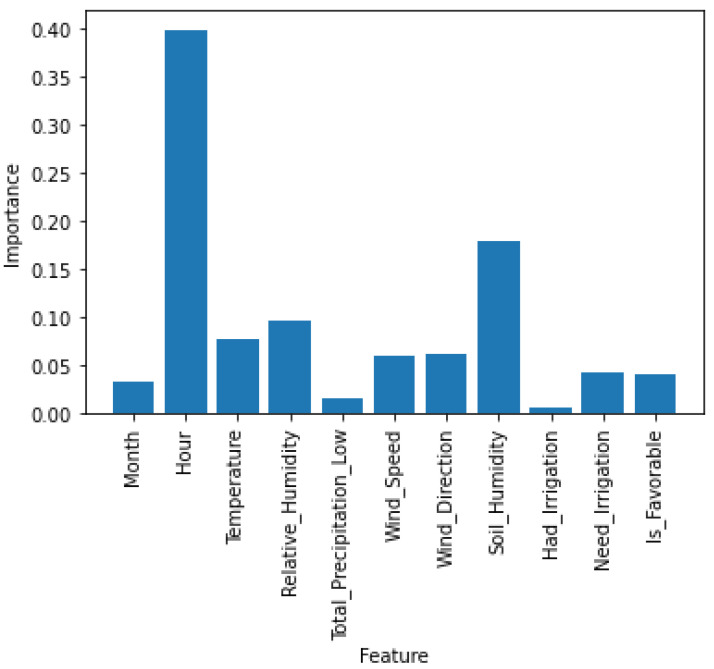
Parameters Analysis.

**Figure 5 sensors-21-03079-f005:**
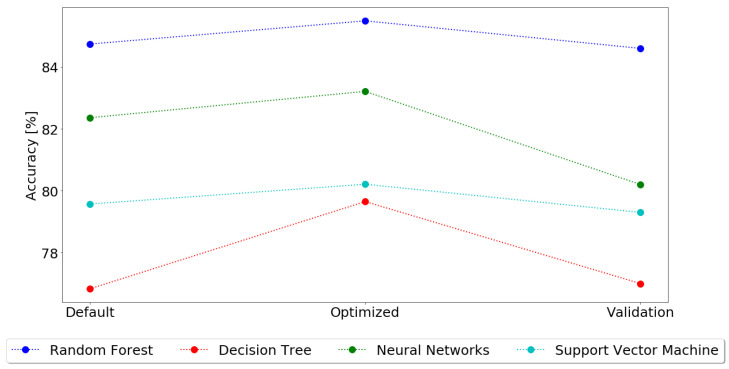
Classification Results.

**Figure 6 sensors-21-03079-f006:**
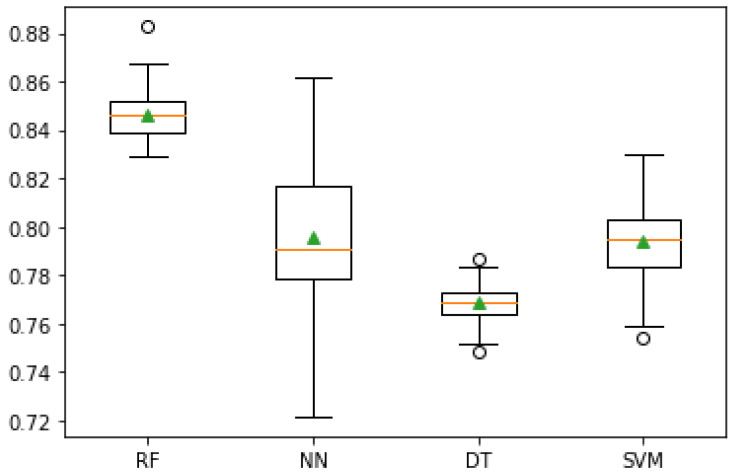
Mean Accuracy after Cross-Validation.

**Figure 7 sensors-21-03079-f007:**
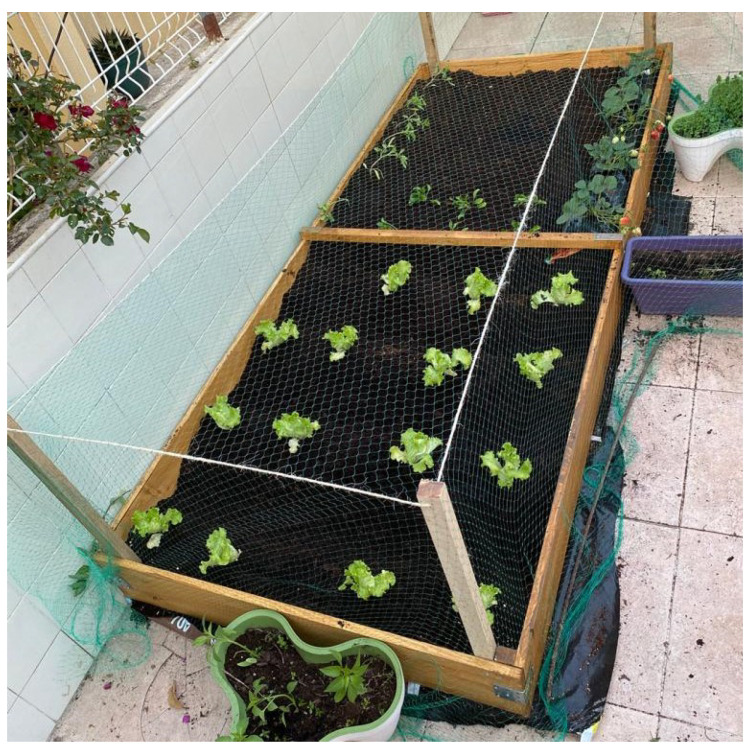
Tested Urban Farm.

**Figure 8 sensors-21-03079-f008:**
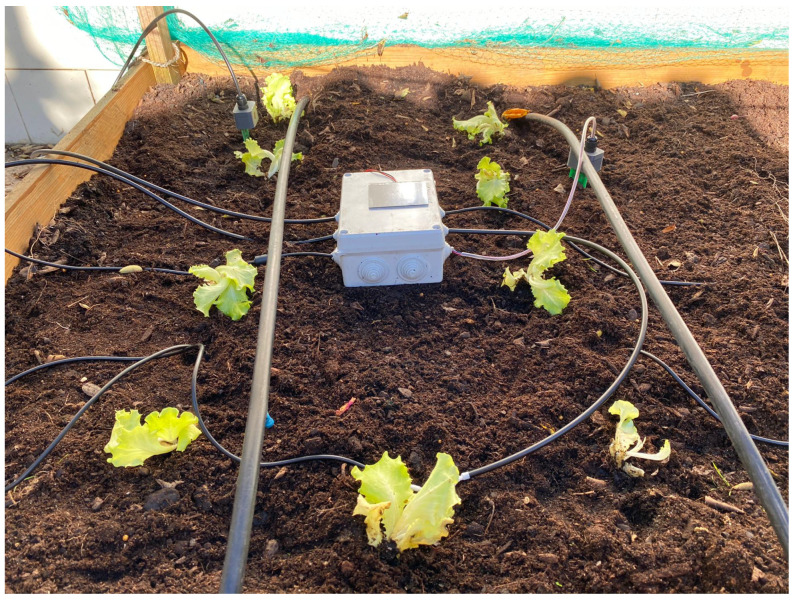
System Implementation—Sensor Nodes.

**Figure 9 sensors-21-03079-f009:**
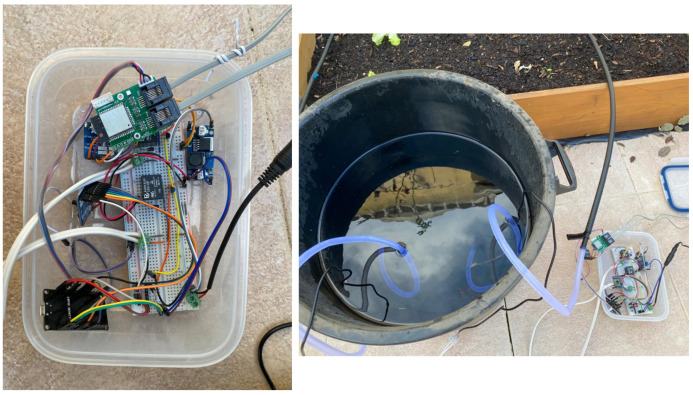
System Implementation—Actuator Node.

**Figure 10 sensors-21-03079-f010:**
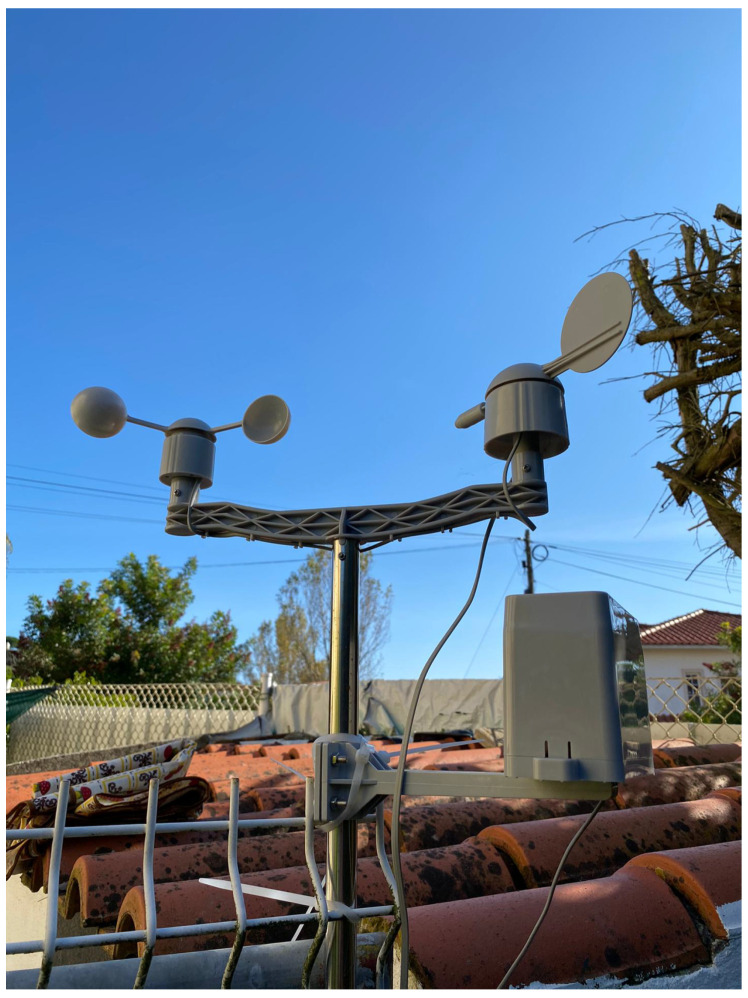
Weather Station.

**Figure 11 sensors-21-03079-f011:**
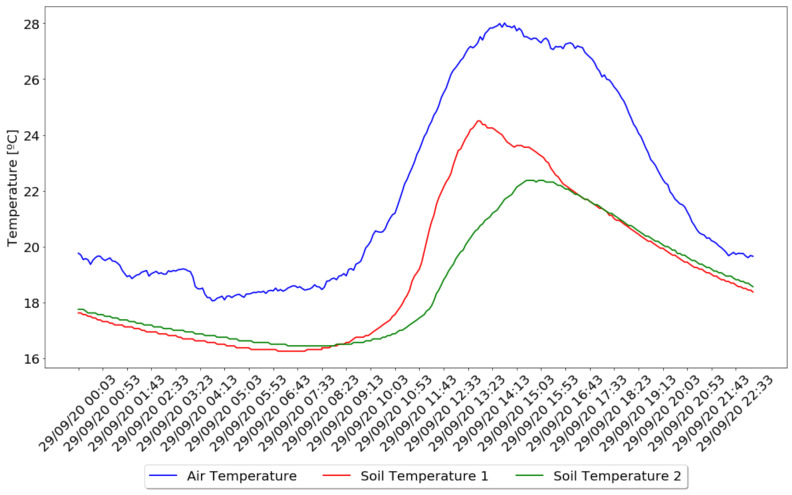
Air and Soil Temperature Comparison.

**Figure 12 sensors-21-03079-f012:**
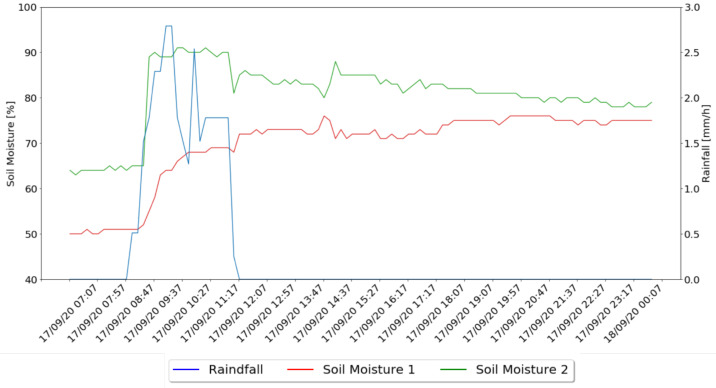
Soil Moisture Variation During Rain.

**Figure 13 sensors-21-03079-f013:**
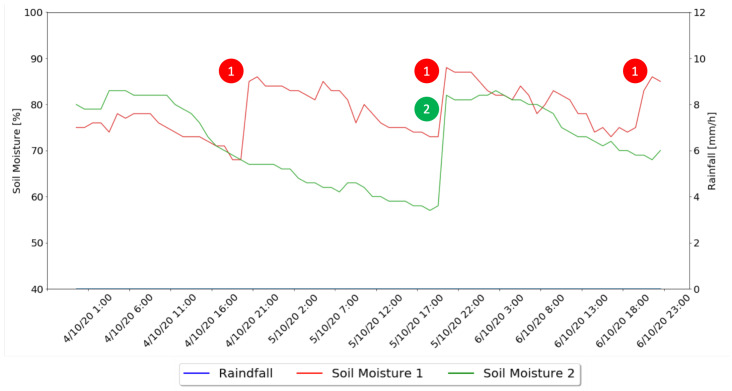
Sunny Days Results Graph.

**Figure 14 sensors-21-03079-f014:**
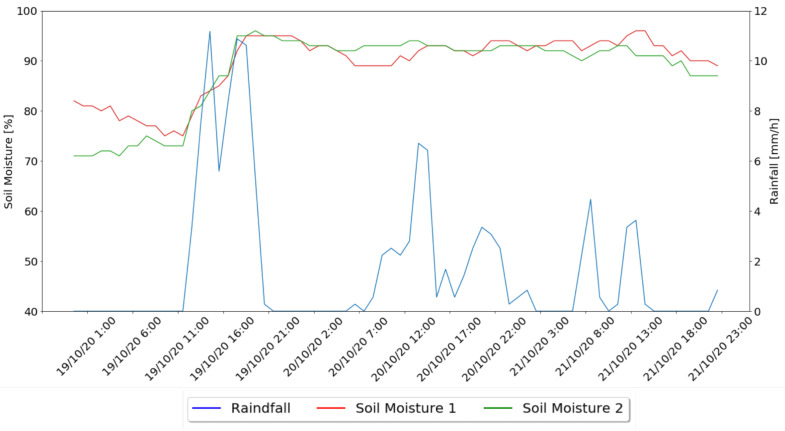
Rainy Days Results Graph.

**Figure 15 sensors-21-03079-f015:**
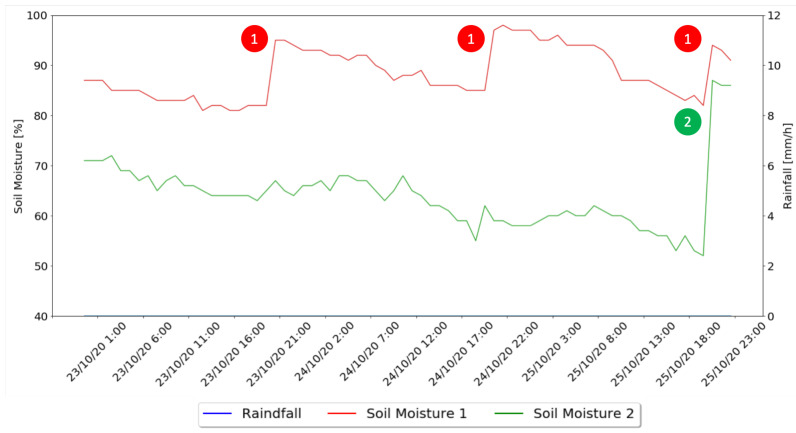
Days After Rainy Days Graph.

**Table 1 sensors-21-03079-t001:** Average sensor data vs API weather data.

Feature	Sensor Data	API Data	Variance
Air Temperature	14.28	13.93	+0.35 ${}^{{}^{\circ}}$C
Air Humidity	68.22	72.76	−1.54%
Precipitation	0.18	0.22	−1.4 mm/h
Wind Speed	11.8	11.5	+0.3 km/h
Wind Direction	201.23	199.64	+1.59${}^{{}^{\circ}}$

**Table 2 sensors-21-03079-t002:** Dataset Summary.

Feature	Count	Mean	Standard Deviation	Minimum	Maximum
Year	105,217	2017	1.95	2014	2020
Month	105,217	6	3.44	1	12
Day	105,217	15	8.8	1	31
Hour	105,217	12	6.92	0	23
Temperature	105,217	13.09	8.01	−1.79	40.37
Relative_Humidity	105,217	70.57	15.06	17.0	110.0
Total_Precipitation_Low	105,217	0.1	0.34	0.0	8.9
Wind_Speed	105,217	11.01	8.22	0	80.0
Wind_Direction	105,217	199.91	92.41	0.8	360.0
Soil_Humidity	105,217	72.89	21.88	5.58	100.0
Had_irrigation	105,217	0.083	0.27	0	1
Need_Irrigation	105,217	0.15	0.35	0	1
Is_Favorable	105,217	0.74	0.43	0	1
Suggested_Hour	105,217	13	5.98	0	23

**Table 3 sensors-21-03079-t003:** Paired Statistical Test Results.

	Decision Trees	Neural Networks	SVM
*p*-value	0.038	0.027	0.032

**Table 4 sensors-21-03079-t004:** Irrigation Times.

Day	Method 1		Method 2	
	Irrigation	Water	Irrigation	Water
	Time (s)	Usage (l)	Time (s)	Usage (l)
04/10	32	3.11	-	-
05/10	24	2.33	52	5.05
06/10	24	2.33	-	-
19/10	-	-	-	-
20/10	-	-	-	-
21/10	-	-	-	-
23/10	19	1.85	-	-
24/10	16	1.55	-	-
25/10	18	1.75	49	4.76

**Table 5 sensors-21-03079-t005:** Water Usage.

Test	Days	Avg. Water			Total		
		Used per Day [l]			Consumption [l]		
		1	2	3	1	2	3
**Sunny**	45	3.12	2.33	5.75	140.4	104.85	258.75
**Rainy**	28	0	0	0	0	0	0
**After Rain**	17	2.17	2.22	5.75	36.89	37.74	97.75
**Total**	90				177.29	142.59	356.5

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
