# Peer review of "Sustainable Irrigation System for Farming Supported by Machine Learning and Real-Time Sensor Data â€"

_sensors, 2021, doi:10.3390/s21093079_

Round 1
Reviewer 1 Report
I read the paper titled "Sustainable Irrigation System for Farming Supported by Machine Learning and Real-Time Sensor Data". I found the research interesting. In fact, although the topic is well known, the results are interesting because provide a new approach to use for use of Machine Learning techniques, that includes the training methodology and a comprehensive comparison of this techniques for irrigation purposes will be shown. The paper is well written and the goal is focused and well-realized. The manuscript is potentially interesting for the readers of Sensors. As a reviewer, I still have some comments and suggestions.
- The results were not at all compared with the research of other authors. It is difficult to evaluate the correctness of the experiments and the results without comparison. It is necessary to find as close as possible research oriented in terms of materials and parameters, because in the present form it is only the presentation of the results.
- All cited sources concern only to Introduction part, none are assigned to the discussion or comparison of the results. Add more references with similar issues and focus of journal Sensors.
- The framework and detailed algorithm for the proposed method should be provided.
- Please provide the statistical test results.
- Please indicate how many rounds of trials were conducted? Please also report their variances.
- Please validate the proposed method on multiple datasets.
Reviewer 2 Report
The proposed manuscript is discussing a case study for an irrigation system with real-time sensor data. The study is practically established and attempted to draw conclusions based on the sensor data. The manuscript is well written but the research contributions seem limited, rather, it looks more like the description of a practical IoT experiment. It becomes very crucial how the authors justify the novelty in terms of research contributions of the manuscript in the revised draft. Besides, the authors must consider the following comments to improve the manuscript. 1. Motivation and research component of the manuscript is not discussed well in the manuscript. Rather, it looks more like an embedded system and IoT case study in practice. 2. There are numerous such case studies available on the web, including several white papers from several Tech companies. To justify the novelty and research contributions, the authors must compare the performance and features of different such systems. 3. The dataset is not well described. The authors must provide statistical characteristics and visualization of used and targeted datasets for a better understanding of the study. 4. The implementation, performance evaluation, and comparison of several machine learning methods used in the study are not well described. The authors can refer to some of the ways and techniques used in prediction model evaluations such as https://www.mdpi.com/1996-1073/13/10/2578 . 5. There are several grammatical and spelling errors in the manuscript. Such as 'In this paper is presented the study' in the Abstract.Author Response
Please see the attachment

Reviewer 3 Report
Machine Learning algorithms were studied to predict the
best time of day for water administration. Of the studied algorithms (Decision Trees, Random Forest, Neural Networks and Support Vectors Machines) the one that obtained the best results was Random Forest, presenting an accuracy of 84.6%.
The current algorithms which currently used are old , I would recommend to use more updated algorithm such as CNN, LSTM, bilstm, 2d-cnn and RNN.
The data pre-processing and feature extraction stage should be explained in details.
The data collection steps should be explained in details.
I would recommend to compare the current approach with state-of-the-art approaches such as:
Zheng, C., Abd-Elrahman, A. and Whitaker, V., 2021. Remote Sensing and Machine Learning in Crop Phenotyping and Management, with an Emphasis on Applications in Strawberry Farming. Remote Sensing, 13(3), p.531.
Zahid, A., Dashtipour, K., Carranza, I.E., Abbas, H., Ren, A., Cumming, D.R., Grant, J.P., Imran, M.A. and Abbasi, Q.H., 2021. Machine Learning Framework for the Detection of Anomalies in Aqueous Solutions Using Terahertz Waves.
Round 2
Reviewer 2 Report
Updates in the manuscript are satisfactory.